# Climate windows of opportunity for plant expansion during the Phanerozoic

Khushboo Gurung [1,2] ✉, Katie J. Field[3], Sarah A. Batterman[4,5,6], Yves Goddéris [7], Yannick Donnadieu [8], Philipp Porada [9], Lyla L. Taylor [3] & Benjamin J. W. Mills [2]

Earth's long-term climate may have profoundly influenced plant evolution. Local climatic factors, including water availability, light, and temperature, play a key role in plant physiology and growth, and have fluctuated substantially over geological time. However, the impact of these key climate variables on global plant biomass across the Phanerozoic has not yet been established. Linking climate and dynamic vegetation modelling, we identify two key 'windows of opportunity' during the Ordovician and Jurassic-Paleogene capable of supporting dramatic expansions of potential plant biomass. These conditions are driven by continental dispersion, paleolatitude of continental area and a lack of glaciation, allowing for an intense hydrological cycle and greater water availability. These windows coincide with the initial expansion of land plants and the later angiosperm radiation. Our findings suggest that the timing and expansion of habitable space for plants played an important role in plant evolution and diversification.

The rise of land plants during the Paleozoic Era (541–251 million years ago; Ma) is thought to have marked a turning point in Earth history, with profound impacts on the planet's surface chemistry and climate[1]. The earliest land plants (embryophytes) are identified in the Ordovician period (485–443 Ma) and are morphologically simple compared to modern vascular plants, being rootless and non-vascular, bearing some similarities to bryophytes[2]. Throughout the Paleozoic, terrestrial flora diversified with vascular plants (tracheophytes) first being recorded during the late Silurian (443–419 Ma) and radiating in the Devonian[3] (419–358 Ma, Fig. 1). Continuous adaptation to the local environment over time drove the evolution of stems, leaves, wood and bark in the late Devonian and early Carboniferous[3]. A later major step in plant evolution was the divergence of the angiosperms (flowering plants), estimated to have occurred between 120–100 Ma based on the occurrence of flowers in the fossil record[4], although angiosperms are predicted to have diverged much earlier than this according to molecular data analysis (Fig. 1). Angiosperms rapidly spread and diversified

due to their high reproductive and growth rates[5], eventually dominating terrestrial plant assemblages throughout the remainder of the Cretaceous[6,7]. The continued success of angiosperms is exemplified in the lowland tropical rainforest of the Neotropics where more than 90% of plant species are angiosperms[8].

Plants likely had dramatic impacts on the composition of the atmosphere by drawing down and photosynthetically fixing atmospheric $CO_2$ into organic biomolecules, and by altering the continental weathering processes which are a key part of most major biogeochemical cycles[9,10]. Through their influence on atmospheric composition and biogeochemical cycles, it has been hypothesised that plants had a key role in driving both the Hirnantian (~445 Ma) and Late Paleozoic (~300 Ma) ice ages[10,11] as well as mid-Paleozoic oxygenation of the atmosphere[12] and the more recent Cenozoic cooling[13,14] (66 Ma – present). However, while the general trajectory of plant evolution is relatively well understood, it remains difficult to estimate changes in global plant biomass, which will affect the magnitude of any impacts

[1]Centre for Plant Sciences, School of Biology, University of Leeds, Leeds LS2 9JT, UK. [2]School of Earth and Environment, University of Leeds, Leeds LS2 9JT, UK. [3]Plants, Photosynthesis and Soil, School of Biosciences, University of Sheffield, Sheffield S10 2TN, UK. [4]Cary Institute of Ecosystem Studies, Millbrook, NY 12545, USA. [5]School of Geography, University of Leeds, Leeds LS2 9JT, UK. [6]Smithsonian Tropical Research Institute, Ancon, Panama. [7]Géosciences Environnement Toulouse, CNRS-Université de Toulouse III, Toulouse, France. [8]CEREGE, Aix Marseille Univ, CNRS, IRD, INRA, Coll France, Aix-en-Provence, France. [9]Institute of Plant Science and Microbiology, University of Hamburg, Hamburg, Germany. ✉e-mail: k.gurung@leeds.ac.uk

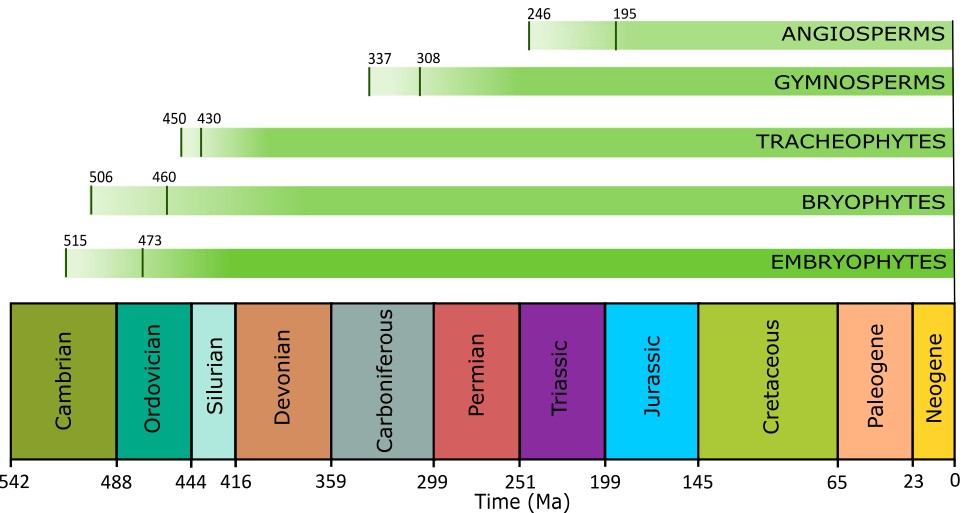

**Fig. 1 | Approximate estimations of plant evolution and Phanerozoic time periods.** Lines indicate the earliest and latest origin of embryophytes, bryophytes, tracheophytes, gymnosperms and angiosperms estimates according to molecular clock analysis[41]. Numbers on the timeline indicate the start of each Phanerozoic period; Ma: million years ago.

on climate and biogeochemical cycles. The methods of quantification and modelling of the land biosphere in the above cited work tend to rely on either box modelling (i.e. non-dimensional models that predict global averages with no spatial information), with no consideration of local hydrology and the impact of water availability on key plant physiological processes[10,11,15], or on complex spatial vegetation models which are set up for specific time periods and are not easily extended across Phanerozoic time[12,16] (541 Ma – present).

Earth's paleogeography is a key feature that regulates plant productivity and biomass at the global scale as it modulates local hydrology and temperature[17,18]. Conditions dictated by changes in paleogeography therefore can enhance or diminish plant growth and could have been a key factor in the expansion of new plant groups and species[19]. One of the biggest changes in paleogeography during the Phanerozoic was the breakup of the supercontinent Pangea (Fig. S1) which saw the transition away from an Earth surface where runoff was limited due to the reduction of inland rainfall[20]. The breakup of the supercontinent and the subsequent enhancement of the hydrological cycle via the formation of a new ocean[21] may have led to the expansion of temperate zones and introduced new niches which could have promoted angiosperm radiation during the Cretaceous[18] (145–66 Ma). Despite these important hypotheses, there has been relatively little exploration of global biomass under past climates. Detailed studies of past vegetation dynamics generally explore the evolution of plant distribution and diversity, and periodical changes in ecology[3,22,23] but very few explore the magnitude of change in global plant productivity, especially over millions of years. The most progress to date on quantifying paleo-biomass has been achieved through use of the Sheffield Dynamic Global Vegetation Model (SDGVM) but this, and similar studies, have been restricted to post-Pangaea climates[16,24].

Here, we develop a simplified deep-time dynamic global vegetation model which can easily be run for a variety of past climates throughout the Phanerozoic to test the hypothesis that paleogeography itself has influenced the spread of plants across Earth's terrestrial land masses. We validate our model against present day distribution of plant biomass and the previous SDGVM work, and explore the effect of Phanerozoic continental dispersion, temperature and runoff on the potential for the Earth to host plant biomass. Our model does not explicitly consider the ability of plants to modify global biogeochemical cycles, and does not include evolutionary differentiation and radiation. However, our results for global plant biomass act to inform thinking on these important aspects.

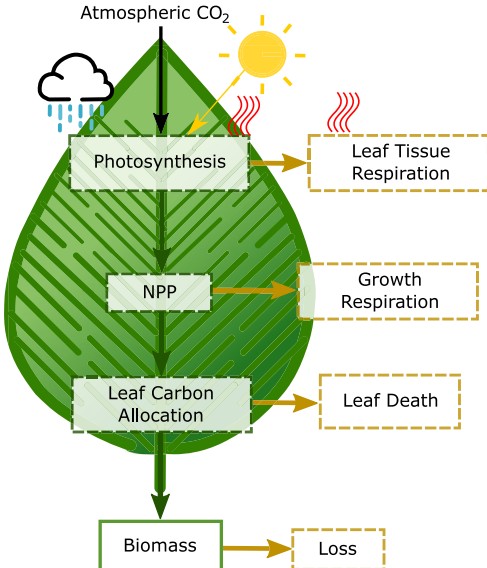

**Fig. 2 | Model flowchart.** Each arrow depicts the flow of carbon, green indicates carbon is preserved within the system while brown indicates its departure. Note: the model uses a single biomass pool and losses associated with respiration and leaf death affect the growth of the biomass pool. Processes are given in dashed boxes whereas reservoirs are presented in bold boxes. Processes that are affected by temperature (red lines), insolation (arrow from sun) and water stress (blue rain) are indicated. NPP: net primary productivity.

Our deep-time vegetation model is called FLORA: Fast Land Occupancy and Reaction Algorithm. This acronym embodies the key considerations of the model; computational speed, the ability to determine if each land grid cell in a climate model is suitable for plant growth, providing an estimate of the total productivity and biomass for each cell through modelling the photosynthesis and respiration reactions. Our model is largely simplified from the Lund-Potsdam-Jena DGVM (LPJ-DGVM)[25] and captures the flow of carbon from its atmospheric form ($CO_2$) to storage as biomass in plants (Fig. 2). The main processes of photosynthesis, mortality and growth rates have been taken from the LPJ-DGVM with simplifications to reduce plant functional types and carbon reservoir types (i.e. removing explicit

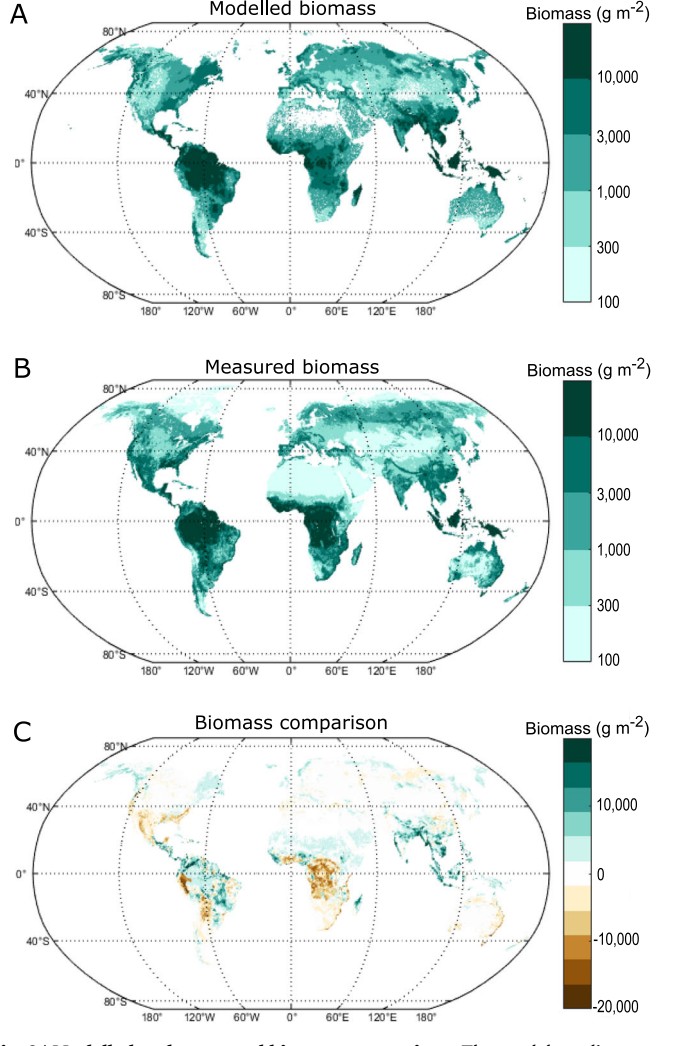

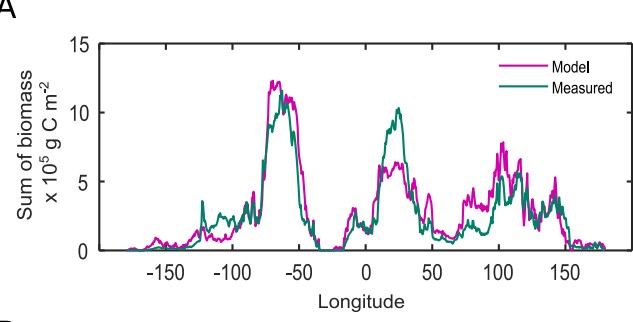

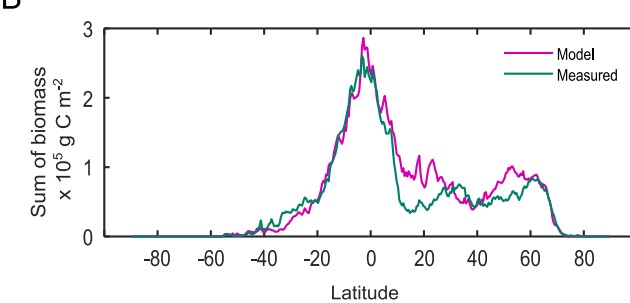

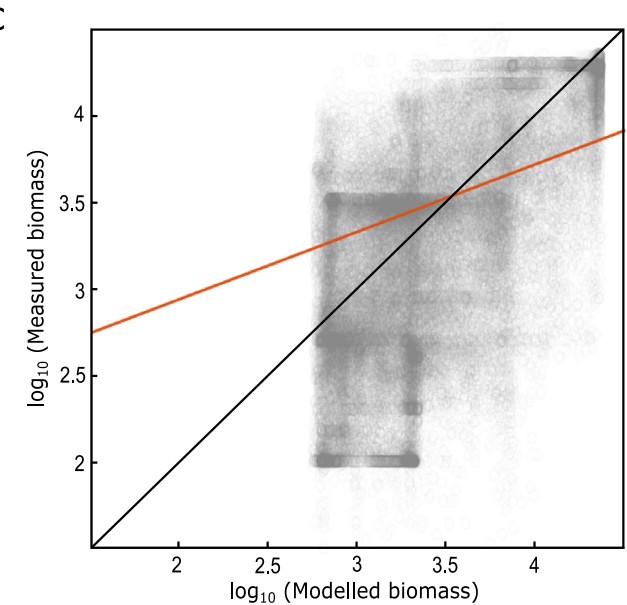

**Fig. 3 | Modelled and measured biomass comparison.** The model predicts a reasonable approximation of current biomass. **A** Model predicted biomass given average temperatures (between 1900–1990) and 'best estimation' of yearly runoff from the year 2000. **B** Actual above- and below-ground global biomass for the year 2000 obtained from CDIAC[35]. **C** Areas of over-prediction (green) and under-prediction (brown) of biomass.

**Fig. 4 | Further modelled and measured biomass comparison. A**, **B** Sum of biomass between modelled (pink line) and measured (green line) data[35] show overall longitude (−180°W to 180°E) and latitude (−90°S to 90°N) biomass patterns are preserved. Highest global plant biomass is present closer to the equator (0° Latitude). **C** Model predicted and measured biomass show a linear relationship with an R-squared value of 0.332 in log space (orange line), or 0.496 in linear space. A 1:1 line is shown for comparison in black.

treatment of sapwood and roots), and related plant processes such as carbon allocation. Processes involving larger ecological interactions such as canopy cover, fire, soil structure, and establishment rates have also been excluded, reducing each grid cell to a simplified metabolism capable of photosynthesis and respiration in order to determine productivity rates and overall biomass with minimum computational requirement. The advantage of this simplification is that FLORA can be run very quickly and in-line with larger biogeochemical frameworks while retaining similar predictions of vegetation carbon distribution to those of the LPJ-DGVM.

We define Net Primary Productivity (NPP) as the net carbon stored after autotrophic respiration[26]. We assume that all plant carbon within the system is stored in the form of leaf biomass for simplicity. Although root and sapwood biomass are present within the LPJ-DGVM[25], they are closely linked to the other biomass pools and are not required to reproduce a reasonable fit to modern biomass (see validation below). Moreover, these features were absent from early plants, thus we opt for the simplest approach. The methods section outlines the equations that dictate the rate of photosynthesis and respiration, carbon allocation and turnover as a response to local solar insolation, temperature and water availability, as well as to the

atmospheric $CO_2$ and $O_2$ levels. Despite their presence in the late Carboniferous as arborescent lycophytes[27], crassulacean acid metabolism (CAM) plants are excluded from the model at its present state due to the lack of research on CAM modelling and absence from vegetation models including the LPJ[28]. C4 photosynthesis was also excluded due to its lack of dominance before the late Miocene (5.3–8 Ma)[29]. Plants are modelled according to C3 photosynthesis as it is the ancestral pathway for carbon fixation and occurs in all taxonomic plant groups[30].

Carbon flows and biomass are calculated over a grid of cells representing the continental surface, and for three basic plant functional types: tropical, boreal and temperate. The only distinction

between each plant functional type is their performance at different temperatures[31]; each plant functional type has a different optimum temperature for photosynthesis (Table S1). A simple competition model for each grid cell allows only the contribution of the functional type for which the highest potential biomass is calculated, thus dictating the 'biome' of the grid cell.

We ran FLORA subject to boundary conditions of the pre-industrial $CO_2$ and $O_2$ levels, 0.5 degree gridded global runoff[32] and temperature[33] measurements, and a standardised insolation curve peaking at 400 W m$^{-2}$ at the equator[34]. Despite the simplicity of the model, the predicted global pattern of biomass shows good agreement with the measured global biomass[35] (Fig. 3). The largest errors occur in the tropics but vary from over-prediction in South Asia and Indonesia, to under-prediction in tropical Africa. The maximum error in a single grid cell is about $2 \times 10^4$ gC m$^{-2}$, but errors tend to be balanced when considering larger areas and the overall pattern in biomass (Fig. 4). Our model also tends to slightly over-represent biomass in the northern high mid-latitudes and under-represent biomass in the southern high mid-latitudes. Such under-representation is possibly due to the absence of plant functional types that act as gradients between biomes and our restriction of a single best-adapted and non-evolving functional type to each grid-cell. Additionally, existing biases present within such complex vegetation models (such as the LPJ-DGVM) may have been inherited. These include, but are not limited to, overestimations of the maximum carboxylation rate between plant functional types[36] and with changing $CO_2$ concentrations[37]. Overall, the biomes in the model follow a similar geographic range to those on the present day Earth (Fig. S5).

Figure 4 shows the longitudinal and latitudinal biomass comparisons and the relationship between the model and the global database[35]. These highlight that the model has reasonable capabilities in capturing the key trends, and again show the slight over-prediction of biomass in South-East Asia and under-prediction in tropical Africa. These differences may be attributable to the yearly-averaged datasets that are used as forcings. For example, seasonal changes in runoff and productivity are not captured which means monsoonal climates are not well-represented. Overall, we consider the model to be appropriate to the task it is designed for. It suitably reconstructs the major patterns of plant biomass on the present-day Earth.

## Results and discussion

### Potential plant biomass over the Phanerozoic

FLORA was run for the paleogeography, surface air temperature and runoff calculated by Goddéris et al.[20] using the FOAM (Fast Ocean Atmosphere Model[38]) climate model for 22 'snapshot' time points over the Phanerozoic. $CO_2$ concentration for each run was set based on proxy information or box modelling where proxies are unavailable (Table S2), and we assume a linear increase in insolation over time (see methods). For these runs we do not consider any evolutionary changes in the land biosphere, thus our calculation is for 'potential biomass' under our generalised photosynthesis-respiration model with modern plant functional types. The intention here is to understand the biomass potential of past climates based on fundamental photosynthetic processes and parameters. Our results are shown in Fig. 5 and Fig. 6 alongside model parameters: area, runoff, $CO_2$ level and average temperature of each climate model run. These results indicate two clear peaks in potential biomass; the first being during the Ordovician, and the second being a broader peak from the Jurassic to the Paleogene (201 Ma–23 Ma) (Fig. 6A). The same pattern is observed in NPP due to the allocation of productivity only towards biomass (Fig. S2).

The peaks in potential biomass predicted by our model are consistent with times of generally elevated global runoff (Fig. 6A, D), which is understandable given the absolute requirement for water for plant growth. Using linear regression between global biomass and global environmental parameters for each grid cell, runoff has the highest

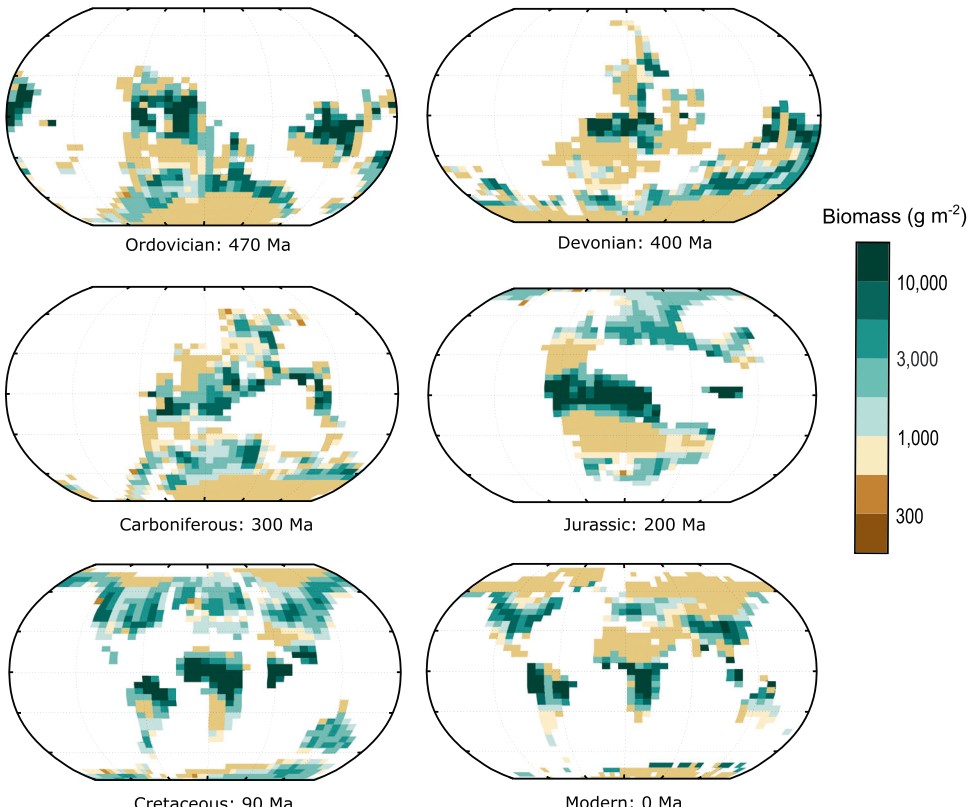

**Fig. 5 | Global potential biomass maps during the Phanerozoic.** Maps showing a selection of the potential biomass predictions from this model. Ma: million years ago.

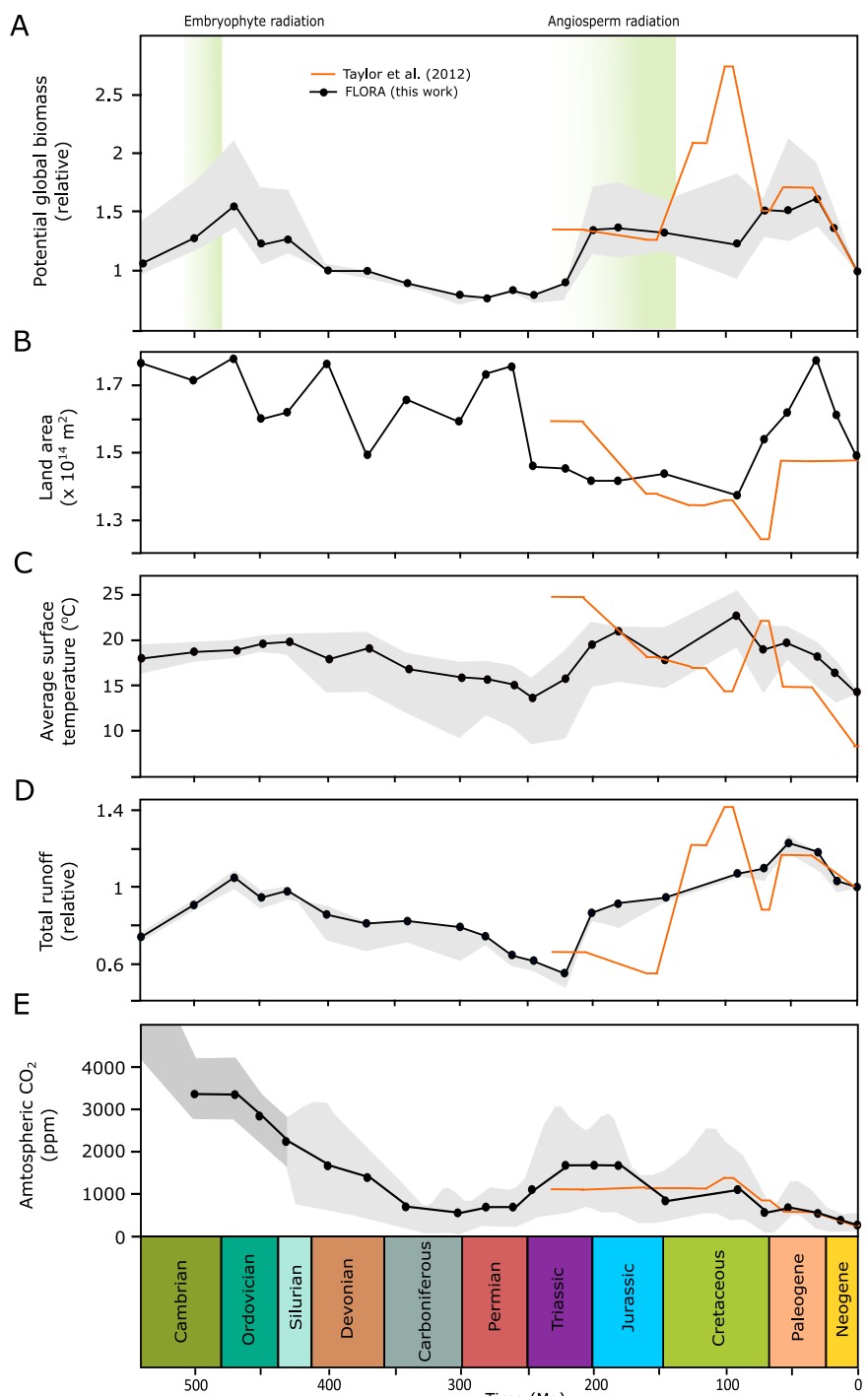

**Fig. 6 | Global potential relative biomass, runoff, average temperature and $CO_2$ level during the Phanerozoic (540–0 Ma).** Temperature and runoff depend on the predicted $CO_2$ level. Grey shaded area represents the min/max values obtained at the min/max $CO_2$ level for each panel apart from E. Time periods are highlighted at the bottom of the figure. Parameters and biomass used in Taylor et al.[16] are shown in orange; the length of the solid line represents time periods used. Ma: million years ago (**A**) Relative biomass over time (kg C relative to present). Green highlights show predicted embryophyte and angiosperm radiation[4,42]. **B** Total terrestrial land area (m²) present at each time point. **C**, **D** Average surface temperature (°C) and relative runoff, respectively. Taken from Goddéris et al.[20] **E** Average $CO_2$ (ppm). Light grey area: an approximate $CO_2$ value was chosen for time periods between 430–0 Ma using Foster et al.[55]. Dark grey area: approximate $CO_2$ values were taken from GEOCARBSULF and COPSE model predictions taken from Mills et al.[56].

r-squared value at individual grid points (average = 0.54; Fig. S3) and as a global value across time (0.62; Fig. S4) suggesting potential biomass most influenced by water availability. There is also some correlation between potential biomass with temperature, where the expansion of ice caps during the late Paleozoic and the late Cenozoic limited the habitable space, but very warm climates like the late Cretaceous also limited productivity. Our findings compare well to those of Taylor et al.[16], who coupled the SDGVM to the Hadley Centre general circulation model (HadCM3L) for a more limited set of paleoclimates. They also found high reconstructed global biomass across the Cretaceous and Paleogene (145–23 Ma). The large disparity seen in the Cenomanian appears to be due to differences in the exposed land area in

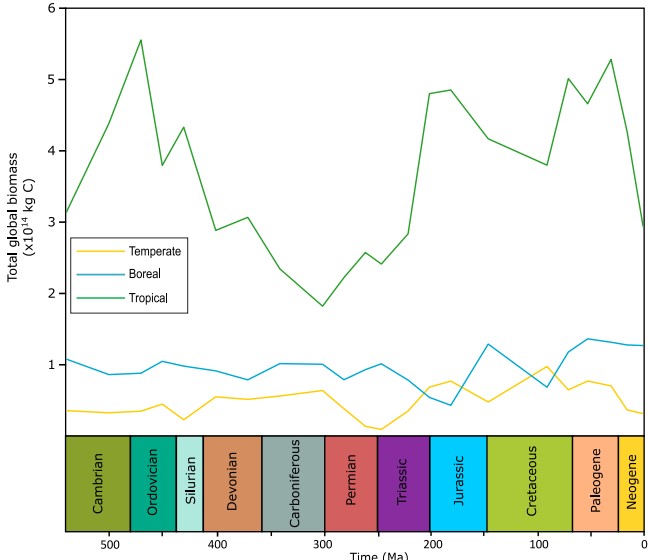

**Fig. 7 | Potential biomass of plant functional types across the Phanerozoic.**
Tropical plant functional type biomass dominates across the Phanerozoic and is the driver for the key changes in our global potential biomass predictions. Ma: million years ago.

the tropics between the climate model runs. The modelling of Taylor et al.[16] assumes a large exposed African continent in the tropics whereas the reconstruction used in FOAM for this work has much of the continent flooded. Nevertheless, both the previous work, and our analysis agree that the breakup of Pangaea (Fig. S1) was accompanied by a substantial increase in the habitable space available for plants, most of it corresponding to our tropical biome (Fig. 7).

The tropical biome is responsible for >50% of the total biomass throughout the Phanerozoic with its smallest relative mass occurring over the Carboniferous and Permian. In line with our hypothesis, the formation of Pangaea and the spread of aridity over large areas around the equatorial belt shrunk the tropical biome more so than the boreal and temperate biomes that lie closer to the poles (Fig. 7). Under- and over-estimations of the biome contribution towards biomass may be present due to the lack of overlapping biomes within grid cells.

**Possible links between climate and plant evolution**
Our results show an early peak in potential biomass at around 470 Ma (Fig. 6A) suggesting temperature and water availability were optimal for plant productivity at this time. During this time period there was substantial low-latitude land mass which was sufficiently dispersed to maintain a strong hydrological cycle, continental temperatures were also warm and there were no permanent ice caps. Embryophytes and other morphologically simple plants present during the Ordovician lacked specialised vascular tissues such as roots or stems[2] that are typically associated with water conduction. These early plants likely existed mostly in equilibrium with surrounding air[39] and their distributions were largely restricted to environments of high water availability. Despite many modern bryophytes displaying poikilohydry (the ability to suspend metabolism during limited water availability), the water desiccation tolerance of early land plants remains unknown, and colonisation of more arid, inland environments would require morphological and physiological strategies to prevent plant water loss. According to the FOAM climate model runs, global runoff was increasing between 540–470 Ma (Fig. 6D). This increase in water availability on land would allow for the increasing productivity on land surface during the Ordovician, allowing early plants[39] to persist on land with minimal risk of dehydration. With the geographical spread of

optimal growth conditions, global plant productivity and therefore biomass is likely to have increased in tandem.

In our model, the favourability of the land surface to plant growth decreases throughout the post-Ordovician Paleozoic. Precipitation and runoff decrease markedly as the amalgamation of Pangaea is completed, and the effects of the cooling in the late Paleozoic also reduced the habitable space for plants. Silurian mesofossils indicate the presence of lignified cell walls and tubular structures essential for water supply towards the peripheral regions of plants which were further developed towards the Devonian[40]. Tracheophytes evolved between around 450–430 Ma[41] and the evolution of roots also fall between the Silurian-Devonian period, beginning with rhizoid structures and ending with extensive rooting systems[42,43]. Thus, this period of increasing aridity is associated with the circular evolution of morphological and physiological innovations in plants, focused towards water acquisition, transport and retention.

The oldest angiosperm fossil is dated to 136 Ma[44], but molecular clocks suggest the early history of angiosperms is cryptic[45], with diversification potentially as early as 195–246 Ma[41]. In our model, potential biomass shows a significant increase around 200 Ma which is sustained until the Neogene (Fig. 6A). This increase in plant habitability is strongly linked to a large rise in global precipitation and runoff following the breakup of Pangea. During this time, equatorial Pangea transitioned from arid conditions to a 'mega-monsoonal' circulation which has previously been proposed to set the stage for the ecological expansion of flowering plants[18]. The separation of land creates a water cycle in areas that previously were arid[46] and the spread of land around the equator increases the land area experiencing high-moderate temperatures for plant growth. Thus, our work supports the inference of a large expansion of habitable space for plants being linked to the mid-late Mesozoic (201–66 Ma) angiosperm radiation. Additionally, FLORA suggests an expansion of tropical habitat with the transition of Pangea into smaller land areas (Figs. 5, 7) consistent with a tropical origin of angiosperms[47]. After the initial increase in plant productivity, a further increase in potential biomass during the Cretaceous (Fig. 6A) may have facilitated the later radiation of angiosperms[48].

Global plant biomass is controlled by a combination of surface air temperature, hydrology and photosynthetically-active radiation, and our simple model, FLORA, based on these factors can reproduce a fair representation of present-day biomass distribution (Fig. 4). In Earth's past, these factors have changed markedly due to the positioning of the continents and changes in radiative forcing. When we run FLORA under the FOAM climate model outputs, we find two clear peaks in the 'potential biomass' – a measure of the Earth surface's ability to host plant life. This analysis shows a strong environmental incentive for plant expansion during the Ordovician and a later window during the Jurassic-to-Paleogene, which correspond with the initial land colonisation and the major radiation of Angiosperms respectively. Moreover, the Silurian-Devonian saw increasing aridity, correlating with a succession of plant adaptions in favour of water transport and retention. We propose that these windows of opportunity played a key part in initiating these evolutionary expansions.

## Methods
### Model equations
Most equations are directly taken or slightly altered from Eq. 1-25 in Sitch et al.[25] and Eq. 4-26 from Haxeltine & Prentice[49]. Photosynthesis rate, $P$ (gC m$^{-2}$ year$^{-1}$) is given by:

$$P = 3650\,ins\left(\frac{c_1}{c_2}\right)\left[c_2 - (2\theta - 1)s - 2(c_2 - \theta s)\sigma_c\right]\omega \qquad (1)$$

where photosynthesis is scaled by water stress, $\omega$ and insolation, $ins$. $\omega$ is calculated as a fraction of runoff that ranges from 0–1; 0 being no

water available and 1 being maximum water availability for plants. $ins$ is assumed to be in a linear relationship with latitude, $f_{lat}$[34].

$$ins_0 = 150 + 250 f_{lat} \quad (2)$$

$$ins = ins_0 - ins_0 \times 0.046 \left( \frac{t}{570} \right) \quad (3)$$

Insolation increases as latitudes get closer to the equator and decrease as they go towards the poles. Present day insolation, $ins_0$, decreases linearly over time, $t$, in strength by 4.6% until 570 Ma[50]. The model substitutes PAR (Photosynthetically Active Radiation) for insolation. $\sigma_c, s, c_1, c_2$ are taken from Sitch et al.[25] and written as:

$$\sigma_c \left[ 1 - \frac{(c_2 - s)}{(c_2 - \theta s)} \right]^{0.5} \quad (4)$$

$$c_1 = \alpha f_{temp} \frac{(p_i - \Gamma_*)}{(p_i - 2\Gamma_*)} \quad (5)$$

$$c_2 = \frac{(p_i - \Gamma_*)}{\left( p_i - K_C \left( 1 + \frac{pO_2}{K_O} \right) \right)} \quad (6)$$

$$s = \left( \frac{24}{h} \right) a \quad (7)$$

where $\alpha$ is the effective ecosystem-level quantum efficiency; $\theta$ is the shape parameter that specifies the degree of co-limitation by light and Rubisco activity; $h$ is the daylight hours which for model simplicity is considered to be 24. $f_{temp}$ is a plant-type specific temperature function that limits photosynthesis at high and low temperatures (Table S1). $\Gamma_*$ is the $CO_2$ compensation point given by:

$$\Gamma_* = \frac{pO_2}{2\tau} \quad (8)$$

where $pO_2$ is the ambient partial pressure of $O_2$ (Pa) taken from Krause et al.[51] (Table S2), and $p_i$ is the intracellular partial pressure of $CO_2$ (Pa) calculated using

$$p_i = \lambda p_a \quad (9)$$

$p_a$, is the ambient partial pressure of $CO_2$ and $\lambda$, a positive parameter that represents the stomata keeping a constant ratio of intracellular to ambient $CO_2$. The ratio ranges from 0.6–0.8 therefore a constant of 0.8 for $C_3$ plants is used[24,46].

Temperature dependent kinetic parameters $K_C, K_O$ and $\tau$ are modelled using

$$k = k_{25} Q_{10}^{0.1(T-25)} \quad (10)$$

taken from Haxeltine and Prentice[49]. $K_C, K_O$ are the Michaelis constants for $CO_2$ and $O_2$ inhibition in the Rubisco reaction. $k_{25}$ is the parameter value at 25 °C and $Q_{10}$ is the relative change in parameter for every 10 °C change in temperature.

Initial carbon allocation to leaves $C_{leaf}$ (gC m$^{-2}$ year$^{-1}$) of the carbon acquired via photosynthesis is calculated using:

$$C_{leaf} = l_{max} P \quad (11)$$

using a leaf carbon allocation ratio, $l_{max}$. Under constant conditions, an allocation ratio of 0.88 is given towards shoot growth[52]. Within angiosperms and gymnosperms, allocation towards leaves has a

maximum of approximately 0.75[53] which decreases with plant growth as more biomass is allocated towards the stems. The maximum value is used throughout therefore assuming 75% of photosynthetic carbon is stored in the leaves. Carbon accumulation overtime is then calculated by:

$$C_{leaf(n+1)} = C_{leaf(n)} \left( 1 - f_{leaf} \right) + l_{max} NPP \quad (12)$$

where $f_{leaf}$ represents leaf turnover for each plant type (Table S1). Initial leaf carbon allocation is calculated using photosynthesis however to calculate leaf carbon accumulation for the global population of plants, $NPP$ is used thereafter.

$$NPP = \left( 1 - R_{growth} \right) \left( P - R_{leaf} \right) \quad (13)$$

NPP is the net primary productivity (gC m$^{-2}$ year$^{-1}$). Carbon is acquired by photosynthesis and lost through growth respiration $R_{growth}$ and maintenance respiration, $R_{leaf}$. 25% of total NPP goes towards $R_{growth}$[24] whereas $R_{leaf}$ is calculated using:

$$R_{leaf} = r \left( \frac{C_{leaf}}{cn_{leaf}} \right) g(T) \quad (14)$$

$$g(T) = \exp \left[ 308.56 \left( \frac{1}{52.02} - \frac{1}{T + 46.02} \right) \right] \quad (15)$$

$R_{leaf}$ depends on a modified Arrhenius equation, $g(T)$, tissue respiration, $r$, and leaf specific C:N ratio, $cn_{leaf}$. $r$ is the plant-type specific respiration rate (gC gN$^{-1}$ year$^{-1}$) (Table S1). Values for the tropical and boreal plant types are taken from Sitch et al.[25] and modified for the temperate plant-type. It follows the observation that plants of warmer environments have a lower respiration rate at any given temperature compared to plants from colder environments[24].

Biomass, $B$, (gC m$^{-2}$ year$^{-1}$) is the reservoir of carbon with inflow from leaf carbon accumulation and we assume a constant 10% outflow, representing combined biomass degradation processes, and chosen to reproduce overall modern biomass.

$$B_{(n+1)} = B_{(n)} + \left( C_{leaf(n)} - 0.1 B_{(n)} \right) \quad (16)$$

Initial biomass $B_{(1)}$ is set at 25 kgC m$^{-2}$ which serves as the baseline for biomass growth/loss.

## Reporting summary

Further information on research design is available in the Nature Research Reporting Summary linked to this article.

## Data availability

All data used to generate biomass in this study has been deposited in the Supplementary Code file. It can also be accessed via GitHub (https://doi.org/10.5281/zenodo.6793631[54]).

## Code availability

The FLORA model is written in MATLAB and is available from KG on request. The code and related material can also be found in the Supplementary Code file or accessed via GitHub (https://doi.org/10.5281/zenodo.6793631[54]).

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

## Acknowledgements

We are grateful to Sitch et al.[25] for the careful description of the equations in the LPJ-DGVM which allowed us to base our model on their work. K.G., K.J.F., S.A.B. and B.J.W.M. are funded by the UK Natural Environment Research Council (NE/S009663/1). K.J.F. is supported by a Philip Leverhulme Prize (PLP-2017-079), a BBSRC Translational Fellowship (BB/M026825/1) and a European Research Council Consolidator Grant (MYCOREV – 865225). S.A.B. is supported by a Philip Leverhulme Prize and UK Nature Environmental Research Council Independent Research Fellowship (NE/M019497/1). P.P.'s contribution to this work was partly funded by Deutsche Forschungsgemeinschaft (DFG, German Research Foundation)—408092731.

## Author contributions

K.G., K.J.F., S.A.B. and B.J.W.M. conceived and designed the investigation. K.G. wrote the model, analysed the data and wrote the first draft of the manuscript with assistance from K.J.F. and B.J.W.M. Y.G., Y.D., P.P. and L.L.T. provided data and discussed results. All authors discussed results and commented on the manuscript. K.G. agrees to serve as the author responsible for correspondence.

## Competing interests

All authors declare no competing interests.
