## [Peer Review File · Nature Communications]

Climate windows of opportunity for plant expansion during the PhanerozoicReviewers' Comments:

Reviewer #1:

Remarks to the Author:

The authors present a dynamic global vegetation model derived and simplified from the Lund-Potsdam-Jena DGVM (LPJ-DGVM) model which they call FLORA, standing for Fast Land Occupancy and Reaction Algorithm. They apply this to a deep-time earth reconstruction at 21 timepoints between 540 Ma and the present. The authors validated their methods by using simulated vs. observed present plant biomass. Using FLORA and past temperature and runoff indexes they show a strong plant biomass expansion during the Ordovician and the Jurassic-to-Paleogene. This is linked with previous suggestions of the major radiation of Angiosperms. They also link the increase in aridity at the Silurian-Devonian with a possible expansion of plant adaptations in favor of water availability related traits. Based on these mix of FLORA simulations with deep-time climate, they propose that these specific events played a key role in the evolutionary expansion of vascular plants.

The manuscript has a good logical flow, and much of the information was presented in a way that made the reading easy to follow. Moreover, the figures were intuitive and easy to grasp. Nevertheless, although the authors mention the strong links between the role of plants in shaping atmospheric gases soil properties (first sentence of the abstract) as well as discuss the radiation and differentiation of plants, these are not addressed by the used methods directly. The authors only provide estimates of biomass changes that might lead to significant macroevolutionary events but biotic contributions to changes in abiotic properties are neglected (or maybe I missed this out). Moreover, by assuming three vegetation types and neglecting the effects of gradual changes and overlaps of multiple plant types, I felt that this manuscript could at least discuss these limitations further. Moreover, there are inconsistencies on how references are presented and how the geological eras are referred (see comments below). With that, I recommend major revisions before this manuscript is accept.

Major comments

The presentation of the results follows an intuitive logic. However, for a manuscript in Nature format, I would try to bring the results presented on figure 5 much earlier in the manuscript. This should highlight the relevance of the work and its findings at a much earlier stage.

At line 47 and 48 the authors mention that Angiosperms continued to diversify throughout the Cretaceous. This should be expanded with proper citations. In the same lines, the authors mention on line 74-76 that most progress has been achieved through use of SDGVM. Thus, the authors neglect several recent developments, specifically some recent studies using realistic eco-evolutionary models that tackle the diversification of plants considering deep-time dynamics and environmental attributes such as temperature and water availability more directly (see for example <https://www.pnas.org/content/pnas/118/40/e2026347118.full.pdf>). Including such studies would enrich the discussion and provide the reader with a better picture of current and future developments in the field, strengthening the aspects of diversification and geodynamics made in the study.

The authors mention that FLORA is largely simplified from the LPJ-DGVM (line 87-88). However, these simplifications were not clearly mentioned. I recommend clearly stating these simplifications either in the text (between lines 84-92) or at Figure 1 legend.

In line with my concerns regarding the simplifications neglecting diversity and biomass production and assuming three vegetation types, could the authors discuss further the assumptions of a constancy in CO₂ conversion across very different physiologies, besides the match between simulated and observed present biomass? If plants adapt and evolve, what could be the effects of assuming constant values in the equations provided? Similarly, on equation 2, is it reasonable to assume constant insolation over such large periods? I recommend either implementing more variable scenarios representing the many present unknowns or making assumptions clearer in the text besides further discussing them.

Minor comments

Some citations are given as number and other as text or are missing. eg. lines 166, 312, 314 as well as COPSE and GEOCARBSULF models. Can you fix these?

I recommend including values for the time periods mentioned to facilitate the reading, as the authors do on line 46. Specifically, at lines: 37, 39, 41, 43, 45, 55, 62, 72, 142, 153 and 200.

Line 57: what do you mean by key factor?

Line 59: what do you mean by box modelling?

Line 88: Potsdam, not Postdam.

Line 172 "environments would" seems like a double space.

Line 177-178: "As the optimal plant growth conditions became more widely distributed global plant productivity is likely have increased in tandem" this sentence is off.

Reviewer #2:

Remarks to the Author:

Manuscript summary:

The authors have constructed a global vegetation model, using past climates to predict concurrent vegetational patterns. Their model predicts considerable variations in potential plant biomass across space and Phanerozoic time, based on runoff, temperature, atmospheric CO₂ levels and habitable area. Although considerable differences between modelled and empirical biomass estimates appear on a grid cell scale when validated with modern data, good agreement is reached when averaging across latitudinal or longitudinal zones. The Phanerozoic estimates of potential plant biomass appear to be most strongly influenced by variations in runoff, largely caused by changes in the continental configuration, with temperature and habitat size playing a secondary role. The authors furthermore propose a link between overall conditions for plant growth and key evolutionary events in plant evolution. Namely, the origin of embryophytes without the means to survive in dry conditions occurred in the early Palaeozoic, at a time of high inferred runoff and pCO₂. The evolution of roots and stems, allowing improved water conduction, happened later in the Palaeozoic, when conditions became drier and less favourable for plant growth, on average. A third event, the radiation of angiosperms inferred by molecular clock estimates, is thought to concur with increasing global potential biomass.

Reviewer conclusion:

The text is well written, concise, and the results are mostly well presented. The manuscript was very interesting to read and understandable for me, being a palaeontologist focussing on marine invertebrates and paleoclimate, not on plants. Estimates of potential biomass across the entire Phanerozoic add an important perspective to the history of plant evolution and improve our understanding of the terrestrial biosphere through time. If published, this manuscript may also enable future work on more detailed predictions of past plant biomass and primary productivity. The results generally support the conclusions drawn by the authors, although some improvements should be made in the statistical analyses and the presentation of the results. If a few shortcomings and minor mistakes (outlined below) are addressed, I support the publication of this paper in Nature Communications.

General comments:

Three plant functional types are included in the model: tropical, boreal and temperate. Tropical

habitats are having the highest potential plant biomass today (Fig. 3b) and throughout the Phanerozoic, judging from Fig. S1. Can the model produce a decomposition into total tropical, boreal and temperate potential biomass across the Phanerozoic (resulting in a plot similar to Fig. 5a, but with three curves)? I suspect that the overall global potential biomass is dominated by the tropical functional types in much of the Phanerozoic. This would be interesting in itself and could allow future work to test hypotheses around the origins and diversity dynamics of plants in different latitudinal zones.

The importance of runoff, temperature, and habitable space ("area") for determining between-grid cell variation in biomass production is determined via linear regression within individual time bins (Fig S2). This should be clarified in the main text (lines 145 – 147), as this was not clear to me without checking the Supplementary Materials. Alternatively, the Methods could be slightly expanded to describe this analysis. Also, it should be mentioned in the figure caption of Fig S2 that the R2 values were obtained in linear regressions. For future works, the authors could consider multiple linear regression (including all independent variables in one regression model).

The role of area in Fig S2 is not clear: Are these regressions done with the land areas of individual grid cells? If so, is this really an informative parameter? A brief comment in the figure caption could help clarify this.

This essentially spatial analysis described above, and shown in Fig. S2, together with visual description of trends in the global averages through time, is used by the authors to argue for the relative importance of environmental factors on overall biomass production through time. The authors should add an additional analysis that quantitatively assesses the relationship of the environmental factors of Fig. 5 on potential global biomass through time. The simplest solution would be a multiple linear regression, using the globally averaged values. If temporal autocorrelation represents a problem, this could be addressed with standard functions (gls in R, I am not familiar with equivalent functions in Matlab).

In the discussion of the angiosperm radiation, it could be added that Cretaceous pollen data indeed indicate a tropical origin of angiosperms (Kvaček, Jiří, et al. "When and Why Nature Gained Angiosperms." *Nature through Time*. Springer, Cham, 2020. 129-158.). Diversification of angiosperms could also have begun a lot later than indicated by molecular clock estimates, which would call into question the link between a mostly early Jurassic increase in plant habitability (Fig. 5) and a possibly much later diversification of angiosperms (Coiro, Mario, James A. Doyle, and Jason Hilton. "How deep is the conflict between molecular and fossil evidence on the age of angiosperms?." *New Phytologist* 223.1 (2019): 83-99.).

In future works, the spatial and temporal analyses could also be integrated into one model, but this is not necessary to support the principal conclusions of this manuscript. Another interesting possibility for future work would be to try and estimate the proportion of potential biomass actually realised in older time periods, when modern plants had not yet evolved.

Specific comments:

Line 88: Spelling mistake in "Potsdam"

Line 91: missing adjective after "very", possibly revise latter part of the sentence

Line 171: poikilohdry, a specialist term deserving explanation

Line 178: is likely to have increased

Line 273: Is 250 kg C correct as the initial biomass? The actual modelled biomass per square meter seems to be in the rage of 100 g to >10,000 g (Fig. 2)

Fig 1: There are no red lines or blue drops (except for the rain cloud). Unclear which processes are affected by temperature, insolation and water stress.

Fig 3: In the figure, there is A, B and C, but in the caption A and B are summarised as A, and C is misspelled as B.

Fig. 5: The orange area around the orange line is very hard to see.

Fig 5e: There is no red dot and no yellow line. Also, the colour of the grey area changes in the Silurian.

Reviewer #3:

Remarks to the Author:

The text is good, clear, didactic and attractive to the reader. The article is very interesting and this speculation about how carbon sequestration may have been in the past of the planet and how this may have influenced the climate is quite relevant.

However, some points need to be improved before they are ready for publication. I think that important information is still missing for the reader.

1. The authors derived the Lund-Postdam-Jena DGVM (LPJ-DGVM) model, the FLORA model and ran it for 541 million years (from the Ordovician period to modern times) estimating the carbon assimilation in the photosynthesis process, the Net Primary Productivity (NPP) and biomass - only in plant leaves. One of the objectives was to test the hypothesis that paleogeography itself influenced the dissemination of plants on Earth and for this, they observed plant biomass in relation to hydrology (runoff), local temperature and the availability of area (surface).

2. The text is good, clear, didactic and attractive to the reader. The article is very interesting and this speculation about how carbon sequestration may have been in the past of the planet and how this may have influenced the climate is quite relevant.

3. However, some points need to be improved before they are ready for publication. I think that important information is still missing for the reader.

4. The authors sought to validate FLORA with results from the SDGVM model coupled to the general atmospheric circulation model (HadCM3L) (Taylor et al. 2012). The ideal would be to run more than one model (DGVM), in addition to FLORA, to give more robustness to the results. They argue that FLORA having only functions and variables runs quite quickly.

Despite 541 million years in the simulation, on line 133, the authors say climate model for 22 time points over the Phanerozoic - which is a small number of points for such a large period. Is that what I understood? And with a low number of times, it doesn't seem to require a lot of computational resources.

5. Not every reader has in mind memorized Earth's geological periods and plant characteristics for each period. I think it would greatly enrich the paper if information such as "The earliest land plants (embryophytes) are first identified in the Ordovician" or "Paleozoic, terrestrial flora diversified with vascular plants (tracheophytes)" could be synthesized in a table that has columns of time (560 MA, 450 MA...), from the Phanerozoic periods (Ordovician, Devonian, Carboniferous, Jurassic, Cretaceous, Modern), from the plant (embryophytes, angiosperms), and main characteristic of this plant (examples: vascular plants,). This would greatly help the reader.

Instead of a table it could also be a figure. In the articles below, I give two examples:

a. Palaeozoic landscapes shaped by plant evolution

Martin R. Gibling and Neil S. Davies, NATURE GEOSCIENCE | VOL 5 | FEBRUARY 2012, a Figure 1 é um exemplo.

b. MUDDYING THE WATERS: MODELING THE EFFECTS OF EARLY LAND PLANTS IN PALEOZOIC ESTUARIES MURIEL Z.M. BRUCKNER, WILLIAM J. MCMAHON AND MAARTEN G. KLEINHANS, the upper part of Figure 1, is interesting, and has information on the evolution of plants over time.

These are examples, but they do not need to be strictly followed, but which can be adapted to the article.

6. An important point is the figures that have a low quality. The delineation of the surfaces in the maps of land are very tenuous, weak, cloudy, poorly defined, they should be sharper and could also contain more information for the reader.

For example, I think Figure 5 is very important in this article. However, I think it could contain more information. One of the information would be to add in the x axis of time, the periods: Ordovician, Devonian, etc...

In the same Figure 5, there could be one more box with the biomass results for each of the Plant Functional Types (PFTs) defined in the simulation: tropical, boreal and temperate.

7. Still, the authors cite a lot of NPP, but do not show results. Could have a figure for NPP, similar to Figure 5 for biomass in Supplementary Information. Ordovician, Devonian, etc...

8. In conclusion:

In lines 207-208, when the authors say:

"In Earth's past, these factors have changed markedly due to the positioning of the continents and changes in radiative forcing".

The simulations with FLORA did not represent the solar variation (standardized insolation curve peaking at 400 W/m^2 at the equator - line 111), so I think this sentence in the conclusion is inappropriate and confuses the reader. It could be in the introduction, but warning that for simplification it was not included in the simulation.

Reviewer response for

Climate windows of opportunity for plant expansion during the Phanerozoic

K. Gurung et al.

We are grateful to the reviewers for their positive and constructive comments. The following document shows these comments in black and adds our responses in blue. We have indicated lines corresponding to each comment in the revised manuscript and altered text in the revised manuscript also appears in blue. Additional figures have been added as suggested by the reviewers in both the supplementary and main text. In addition to changes in response to review, we have extended the paper abstract because the final part was cut off during file conversion, and have updated figure 4c because our initial submission used a slightly out-of-date version with very minor differences.

Reviewer #1

The authors present a dynamic global vegetation model derived and simplified from the Lund-Potsdam-Jena DGVM (LPJ-DGVM) model which they call FLORA, standing for Fast Land Occupancy and Reaction Algorithm. They apply this to a deep-time earth reconstruction at 21 timepoints between 540 Ma and the present. The authors validated their methods by using simulated vs. observed present plant biomass. Using FLORA and past temperature and runoff indexes they show a strong plant biomass expansion during the Ordovician and the Jurassic-to-Paleogene. This is linked with previous suggestions of the major radiation of Angiosperms. They also link the increase in aridity at the Silurian-Devonian with a possible expansion of plant adaptations in favor of water availability related traits. Based on these mix of FLORA simulations with deep-time climate, they propose that these specific events played a key role in the evolutionary expansion of vascular plants.

The manuscript has a good logical flow, and much of the information was presented in a way that made the reading easy to follow. Moreover, the figures were intuitive and easy to grasp. Nevertheless, although the authors mention the strong links between the role of plants in shaping atmospheric gases soil properties (first sentence of the abstract) as well as discuss the radiation and differentiation of plants, these are not addressed by the used methods directly.

The authors only provide estimates of biomass changes that might lead to significant macroevolutionary events but biotic contributions to changes in abiotic properties are neglected (or maybe I missed this out).

We thank the reviewer for the positive assessments and the constructive remarks here and below. We have added text to the introduction to be clearer that feedback between plants and the abiotic environment, and the processes of radiation and differentiation, are not considered in this study – but our model will allow researchers to investigate these things in later work (line 100).

Moreover, by assuming three vegetation types and neglecting the effects of gradual changes and overlaps of multiple plant types, I felt that this manuscript could at least discuss these limitations further.

This is a good point and we have added text to discuss the limitations here (line 152). We have also included Figure S5 (Supplementary) that compares these biomes to simplified real world data and finds a reasonable match to this.

Moreover, there are inconsistencies on how references are presented and how the geological eras are referred (see comments below). With that, I recommend major revisions before this manuscript is

accept.

Apologies for these errors and thank you for spotting them, they have been corrected.

Major comments

The presentation of the results follows an intuitive logic. However, for a manuscript in Nature format, I would try to bring the results presented on figure 5 much earlier in the manuscript. This should highlight the relevance of the work and its findings at a much earlier stage.

This was an error in our file conversion where we chopped off the final part of the abstract which summarised the main results much earlier in the paper. This has now been restored.

At line 47 and 48 the authors mention that Angiosperms continued to diversify throughout the Cretaceous. This should be expanded with proper citations.

We have added citations to this section (line 57).

In the same lines, the authors mention on line 74-76 that most progress has been achieved through use of SDGVM. Thus, the authors neglect several recent developments, specifically some recent studies using realistic eco-evolutionary models that tackle the diversification of plants considering deep-time dynamics and environmental attributes such as temperature and water availability more directly (see for example <https://www.pnas.org/content/pnas/118/40/e2026347118.full.pdf>). Including such studies would enrich the discussion and provide the reader with a better picture of current and future developments in the field, strengthening the aspects of diversification and geodynamics made in the study.

We have expanded our discussion to include this paper and others on changes in deep-time diversity. We have also re-worded our manuscript to be clearer that the progress we attribute to SDGVM is on the distribution of global biomass specifically, which is not considered in the eco-evolutionary studies as far as we can tell (line 89).

The authors mention that FLORA is largely simplified from the LPJ-DGVM (line 87-88). However, these simplifications were not clearly mentioned. I recommend clearly stating these simplifications either in the text (between lines 84-92) or at Figure 1 legend.

This is a good point and we have made this much clearer in the revised paper (line 110).

In line with my concerns regarding the simplifications neglecting diversity and biomass production and assuming three vegetation types, could the authors discuss further the assumptions of a constancy in CO₂ conversion across very different physiologies, besides the match between simulated and observed present biomass?

We have added text on our assumptions here (line 129) – we have assumed that the whole biosphere uses C₃ photosynthesis. Given that C₄ photosynthesis represents only a small fraction of total plant biomass at present, and was absent for most of the Phanerozoic, we believe this simplification is reasonable given the scope of the work. CAM is also not modelled explicitly within vegetation models.

If plants adapt and evolve, what could be the effects of assuming constant values in the equations provided?

This is a good question and including adaption and evolution in our model is a logical next step to this work, and something we are certainly going to pursue. It is possible that this will allow for more global biomass across our simulations, but we do not see it changing the key conclusions of the paper, which are based on the relative biomass at different times. We have added some text on this to the revised paper (line 152).

Similarly, on equation 2, is it reasonable to assume constant insolation over such large periods? I recommend either implementing more variable scenarios representing the many present unknowns or making assumptions clearer in the text besides further discussing them.

This is a great point and we have revised the model to alter the insolation over time. As per Berner 1993, insolation is assumed to decrease linearly by 4.6% from present day until 570 Ma in Equation 2 and 3. The resulting changes are minor as the change in insolation itself is relatively small.

Minor comments

Some citations are given as number and other as text or are missing. eg. lines 166, 312, 314 as well as COPSE and GEOCARBSULF models. Can you fix these?

Apologies for this, we had some issues converting from the initial draft to the journal submission format. We have resolved these now.

I recommend including values for the time periods mentioned to facilitate the reading, as the authors do on line 46. Specifically, at lines: 37, 39, 41, 43, 45, 55, 62, 72, 142, 153 and 200.

We have added these timeframes as suggested throughout the text. Values are given when each period is introduced. Figure 1 now explicitly shows each time period and their dates. Other figures also state all time periods now.

Line 57: what do you mean by key factor?

Here we meant that global biomass is a key factor for models of climate and biogeochemistry, for example most quantitative approaches (like the GEOCARB and COPSE models) tend to link carbon burial or silicate weathering amplification to the amount of biomass. We have revised the statement to make it clearer (line 66).

Line 59: what do you mean by box modelling?

We mean a non-dimensional model which has no spatial dimensions (i.e., all values are a global average). Box modelling does not allow us to observe regional changes. We have made this clearer (line 732).

Line 88: Potsdam, not Postdam.

Whoops. Thanks for the correction (Line 109).

Line 172 “environments would” seems like a double space.

We have checked for and corrected any double spacing throughout the text.

Line 177-178: “As the optimal plant growth conditions became more widely distributed global plant productivity is likely have increased in tandem” this sentence is off.

We have changed this to “With the geographical spread of optimal plant growth conditions, global plant productivity and therefore biomass is likely to have increased in tandem.” (line 230)

Reviewer #2

Manuscript summary:

The authors have constructed a global vegetation model, using past climates to predict concurrent vegetational patterns. Their model predicts considerable variations in potential plant biomass across space and Phanerozoic time, based on runoff, temperature, atmospheric CO₂ levels and habitable area. Although considerable differences between modelled and empirical biomass estimates appear on a grid cell scale when validated with modern data, good agreement is reached when averaging across latitudinal or longitudinal zones. The Phanerozoic estimates of potential plant biomass appear to be most strongly influenced by variations in runoff, largely caused by changes in the continental configuration, with temperature and habitat size playing a secondary role. The authors furthermore propose a link between overall conditions for plant growth and key evolutionary events in plant evolution. Namely, the origin of embryophytes without the means to survive in dry conditions occurred in the early Palaeozoic, at a time of high inferred runoff and pCO₂. The evolution of roots and stems, allowing improved water conduction, happened later in the Palaeozoic, when conditions became drier and less favourable for plant growth, on average. A third event, the radiation of angiosperms inferred by molecular clock estimates, is thought to concur with increasing global potential biomass.

Reviewer conclusion:

The text is well written, concise, and the results are mostly well presented. The manuscript was very interesting to read and understandable for me, being a palaeontologist focussing on marine invertebrates and paleoclimate, not on plants. Estimates of potential biomass across the entire Phanerozoic add an important perspective to the history of plant evolution and improve our understanding of the terrestrial biosphere through time. If published, this manuscript may also enable future work on more detailed predictions of past plant biomass and primary productivity. The results generally support the conclusions drawn by the authors, although some improvements should be made in the statistical analyses and the presentation of the results. If a few shortcomings and minor mistakes (outlined below) are addressed, I support the publication of this paper in Nature Communications.

We thank the reviewer for the positive assessment, and we have revised the paper to address the detailed points below.

General comments:

Three plant functional types are included in the model: tropical, boreal and temperate. Tropical habitats are having the highest potential plant biomass today (Fig. 3b) and throughout the Phanerozoic, judging from Fig. S1. Can the model produce a decomposition into total tropical, boreal and temperate potential biomass across the Phanerozoic (resulting in a plot similar to Fig. 5a, but with three curves)? I suspect that the overall global potential biomass is dominated by the tropical functional types in much of the Phanerozoic. This would be interesting in itself and could allow future work to test hypotheses around the origins and diversity dynamics of plants in different latitudinal zones.

This is a really interesting idea and we have implemented it as suggested. The reviewer is correct that tropical biomass dominates throughout the Phanerozoic in our model and we have added a brief discussion (line 205) along with a new Figure 7 that looks at the plant biomes over time.

The importance of runoff, temperature, and habitable space (“area”) for determining between-grid cell variation in biomass production is determined via linear regression within individual time bins (Fig S2). This should be clarified in the main text (lines 145 – 147), as this was not clear to me without checking the Supplementary Materials. Alternatively, the Methods could be slightly expanded to describe this analysis. Also, it should be mentioned in the figure caption of Fig S2 that the R² values were obtained in linear regressions. For future works, the authors could consider multiple linear regression (including all independent variables in one regression model).

We have made this clearer in the text (line 189) and the supplement. We appreciate the advice to explore a multiple regression in future work, and will attempt this.

The role of area in Fig S2 is not clear: Are these regressions done with the land areas of individual grid cells? If so, is this really an informative parameter? A brief comment in the figure caption could help clarify this.

We have removed this plot as the reviewer is correct that these did use the areas from individual gridcells and therefore the result was not particularly informative.

This essentially spatial analysis described above, and shown in Fig. S2, together with visual description of trends in the global averages through time, is used by the authors to argue for the relative importance of environmental factors on overall biomass production through time. The authors should add an additional analysis that quantitatively assesses the relationship of the environmental factors of Fig. 5 on potential global biomass through time. The simplest solution would be a multiple linear regression, using the globally averaged values. If temporal autocorrelation represents a problem, this could be addressed with standard functions (gls in R, I am not familiar with equivalent functions in Matlab).

This is a great idea and we have added new linear regression analysis between total biomass, and total runoff and temperature as Figure S4 in the SI, and described it beginning at line 189 of the main text. For consistency with the plots already made we have used single regressions.

In the discussion of the angiosperm radiation, it could be added that Cretaceous pollen data indeed indicate a tropical origin of angiosperms (Kvaček, Jiří, et al. "When and Why Nature Gained Angiosperms." *Nature through Time*. Springer, Cham, 2020. 129-158.). Diversification of angiosperms could also have begun a lot later than indicated by molecular clock estimates, which would call into question the link between a mostly early Jurassic increase in plant habitability (Fig. 5) and a possibly much later diversification of angiosperms (Coiro, Mario, James A. Doyle, and Jason Hilton. "How deep is the conflict between molecular and fossil evidence on the age of angiosperms?." *New Phytologist* 223.1 (2019): 83-99.).

We have added this discussion to the paper (Line 255), thank you for the pointers.

In future works, the spatial and temporal analyses could also be integrated into one model, but this is not necessary to support the principal conclusions of this manuscript. Another interesting possibility for future work would be to try and estimate the proportion of potential biomass actually realised in older time periods, when modern plants had not yet evolved.

Yes we agree! We are working on this and hope to find some interesting results.

Specific comments:

Line 88: Spelling mistake in "Potsdam"

Corrected

Line 91: missing adjective after "very", possibly revise latter part of the sentence

Corrected

Line 171: poikilohydry, a specialist term deserving explanation

Brief explanation added alongside

Line 178: is likely to have increased

Corrected

Line 273: Is 250 kg C correct as the initial biomass? The actual modelled biomass per square meter seems to be in the range of 100 g to >10,000 g (Fig. 2)

Apologies this is a typo. The initial biomass is 25 kg C per grid cell. (Line 327)

Fig 1: There are no red lines or blue drops (except for the rain cloud). Unclear which processes are affected by temperature, insolation and water stress.

We have corrected this description and added clearer signs for insolation. This is Figure 2 now.

Fig 3: In the figure, there is A, B and C, but in the caption A and B are summarised as A, and C is misspelled as B.

Corrected

Fig. 5: The orange area around the orange line is very hard to see.

Unfortunately that shading was an unwanted feature of conversion to pdf and is just a blur, we have fixed this. Taylor et al. 2012 did not make error estimates.

Fig 5e: There is no red dot and no yellow line. Also, the colour of the grey area changes in the Silurian.

Thanks for pointing these out. All are corrected now. The change in colour indicated the source of the data (model vs proxy) which is now explained in the caption.

Reviewer #3

1. The authors derived the Lund-Postdam-Jena DGVM (LPJ-DGVM) model, the FLORA model and ran it for 541 million years (from the Ordovician period to modern times) estimating the carbon assimilation in the photosynthesis process, the Net Primary Productivity (NPP) and biomass - only in plant leaves. One of the objectives was to test the hypothesis that paleogeography itself influenced the dissemination of plants on Earth and for this, they observed plant biomass in relation to hydrology (runoff), local temperature and the availability of area (surface).

2. The text is good, clear, didactic and attractive to the reader. The article is very interesting and this speculation about how carbon sequestration may have been in the past of the planet and how this may have influenced the climate is quite relevant.

3. However, some points need to be improved before they are ready for publication. I think that important information is still missing for the reader.

We thank the reviewer for the positive assessment and have addressed the constructive points below.

4. The authors sought to validate FLORA with results from the SDGVM model coupled to the general atmospheric circulation model (HadCM3L) (Taylor et al. 2012). The ideal would be to run more than one model (DGVM), in addition to FLORA, to give more robustness to the results.

What other models out there can we run?

Much of the motivation for this work was the lack of other models available. The SDGVM is the only dynamic vegetation model we are aware of that has estimated biomass over these long timeframes. Study of the present day vegetation uses more complex models, but these tend to take much longer to run and require setting of types and locations of biomes by hand, as well as more complicated soil and ecological properties, meaning they are not well suited to predicting the spread of global biomass over deep time. We have added discussion around line 89 about the current state of the art in vegetation modelling.

They argue that FLORA having only functions and variables runs quite quickly. Despite 541 million years in the simulation, on line 133, the authors say climate model for 22 time points over the Phanerozoic – which is a small number of points for such a large period. Is that what I understood? And with a low number of times, it doesn't seem to require a lot of computational resources.

The snapshots used in the model spans 541 million years. The timepoints depend on paleoclimate data that is available. It also runs into steady state at the moment meaning that the biomass output we get are for those specific snapshots only. We have to interpret the transition between each. Large computational resource is only required with the sensitivity analysis.

Yes this is correct, this paper shows model snapshots for 22 time points, and running each individual model to steady state takes only a minute or so. Nevertheless, developing and testing the model has required thousands of runs, so computational speed was still important. We have added that these runs are 'snapshots' (line 174). We will be performing continuous runs over geological time in future work.

5. Not every reader has in mind memorized Earth's geological periods and plant characteristics for each period. I think it would greatly enrich the paper if information such as “The earliest land plants (embryophytes) are first identified in the Ordovician” or “Paleozoic, terrestrial flora diversified with vascular plants (tracheophytes)” could be synthesized in a table that has columns of time (560 MA, 450 MA...), from the Phanerozoic periods (Ordovician, Devonian, Carboniferous, Jurassic, Cretaceous, Modern), from the plant (embryophytes, angiosperms), and main characteristic of this plant (examples: vascular plants,). This would greatly help the reader. Instead of a table it could also be a figure. In the articles below, I give two examples:

a. Palaeozoic landscapes shaped by plant evolution

Martin R. Gibling and Neil S. Davies, NATURE GEOSCIENCE | VOL 5 | FEBRUARY 2012, a Figure 1 é um exemplo.

b. MUDDYING THE WATERS: MODELLING THE EFFECTS OF EARLY LAND PLANTS IN PALEOZOIC ESTUARIES MURIEL Z.M. BRUCKNER, WILLIAM J. MCMAHON AND MAARTEN G. KLEINHANS, the upper part of Figure 1, is interesting, and has information on the evolution of plants over time.

These are examples, but they do not need to be strictly followed, but which can be adapted to the article.

This is a good idea and we have included a figure like this (Figure 1) in the revision.

6. An important point is the figures that have a low quality. The delineation of the surfaces in the maps of land are very tenuous, weak, cloudy, poorly defined, they should be sharper and could also contain more information for the reader.

Our apologies on the poor quality conversion to pdf. We have used a different converter in the revision.

For example, I think Figure 5 is very important in this article. However, I think it could contain more information. One of the information would be to add in the x axis of time, the periods: Ordovician, Devonian etc...

We have added the full period names to all figures including time.

In the same Figure 5, there could be one more box with the biomass results for each of the Plant Functional Types (PFTs) defined in the simulation: tropical, boreal and temperate.

As per this and a previous reviewer comment, we have added an extra figure for the different PFT biomass overtime and some discussion (line 206; Figure 7).

7. Still, the authors cite a lot of NPP, but do not show results. Could have a figure for NPP, similar to Figure 5 for biomass in Supplementary Information. Ordovician, Devonian etc...

We have included an NPP version of the figure in the SI (Figure S2). It follows the same trends as biomass.

8. In conclusion:

In lines 207-208, when the authors say:

“In Earth’s past, these factors have changed markedly due to the positioning of the continents and changes in radiative forcing”.

The simulations with FLORA did not represent the solar variation (standardized insolation curve peaking at 400 W/m^2 at the equator – line 111), so I think this sentence in the conclusion is inappropriate and confuses the reader. It could be in the introduction, but warning that for simplification it was not included in the simulation.

We have now added explicit changes in solar insolation as it was mentioned by two reviewers. Equations 2-3 implement the decline of radiative forcing for times in the past. Results have also been changed to reflect this although the change is not very large as insolation only decreases by 4.6% during the Phanerozoic.

Reviewers' Comments:

Reviewer #1:

Remarks to the Author:

The authors did a great job replying to the comments and improving the manuscript. I thus, recommend publication.

Reviewer #2:

Remarks to the Author:

The authors have addressed my comments and those of the other reviewers from the first round of reviews. Shortcomings and incomplete descriptions from the initial draft have been remediated.

Therefore, I support the publication of this manuscript, under the condition that a few minor issues in the new version are addressed. Specifically, the following passages need to be improved:

l. 54: Capitalisation of "Early Carboniferous" is a spelling mistake, I assume.

l. 60 – 61: This sentence is misleading! The cited paper states: "early in their evolutionary history, angiosperms could already account for a majority of the species while representing only a small proportion of the ground cover (Wing and Boucher 1998; Wing et al. 1993, 2012), and this is still the case in modern boreal forests, where angiosperms make up at least 90% of the species despite the obvious ecological dominance of conifers (Mutke and Barthlott 2005)."

If you want to bring an example for the success of angiosperm species, it makes much more sense to reference the tropics or the temperate regions, as boreal forests are commonly dominated by conifers.

l. 257: Do you mean that the late Cretaceous increase in potential biomass would have facilitated angiosperm radiation? Phrase more clearly, as now it sounds like the increase may have been caused by angiosperm radiation, which, I presume, cannot be the case, as the model does not know that angiosperms radiated?

Figure caption 7: I appreciate that you included a latitudinal analysis. However, the caption describes a different figure.

Reviewer #3:

Remarks to the Author:

The article is quite interesting and the reading is attractive, as I commented in the first review. The authors fulfilled most of my requests, which greatly improved the manuscript. Some questions that could give more robustness to the results such as run step by step in time, and seek to better represent the transition of vegetation, the article does not contemplate, which is as a suggestion for future works. The comments to the authors' responses follow in an attached file.

In black, reviewer comments in the first round.

In blue, answers from the authors.

In red, reviewer comments in the third round.

1. The authors derived the Lund-Postdam-Jena DGVM (LPJ-DGVM) model, the FLORA model and ran it for 541 million years (from the Ordovician period to modern times) estimating the carbon assimilation in the photosynthesis process, the Net Primary Productivity (NPP) and biomass - only in plant leaves. One of the objectives was to test the hypothesis that paleogeography itself influenced the dissemination of plants on Earth and for this, they observed plant biomass in relation to hydrology (runoff), local temperature and the availability of area (surface).

2. The text is good, clear, didactic and attractive to the reader. The article is very interesting and this speculation about how carbon sequestration may have been in the past of the planet and how this may have influenced the climate is quite relevant.

3. However, some points need to be improved before they are ready for publication. I think that important information is still missing for the reader. We thank the reviewer for the positive assessment and have addressed the constructive points below.

We thank the reviewer for the positive assessment and have addressed the constructive points below.

4. The authors sought to validate FLORA with results from the SDGVM model coupled to the general atmospheric circulation model (HadCM3L) (Taylor et al. 2012). The ideal would be to run more than one model (DGVM), in addition to FLORA, to give more robustness to the results. What other models out there can we run?

Much of the motivation for this work was the lack of other models available. The SDGVM is the only dynamic vegetation model we are aware of that has estimated biomass over these long timeframes. Study of the present day vegetation uses more complex models, but these tend to take much longer to run and require setting of types and locations of biomes by hand, as well as more complicated soil and ecological properties, meaning they are not well suited to predicting the spread of global biomass over deep time. We have added discussion around line 89 about the current state of the art in vegetation modelling.

The explanation of the authors is reasonable. But even so, I still think the work resents the lack of other models for comparison. We know that the models have many uncertainties, mainly in matters of plant physiology. An inaccuracy replicated over time can potentiate the error. In line 89, the authors cite references on vegetation modeling, but these references do not specifically address the uncertainties that occur in DGVMs. Since FLORA, a model derived from LPJ-DGVM, I think it is important that the authors mention that they are aware of these uncertainties and quote more appropriate references to this theme.

Suggestions for references on the uncertainties of the models:

1. Belinda EM et al. (2015). Using ecosystem experiments to improve vegetation models. *Nature Climate Change*. Vol. 5 - June 2015 – doi: 10.1038.

2. Rezende, Luiz F. C. et al., 2022. Impacts of Land Use Change and Atmospheric CO₂ on Gross Primary Productivity (GPP), Evaporation, and Climate in Southern Amazon. *Journal of Geophysical Research: Atmospheres* 10.1029/2021JD034608

3. Rogers, A. (2014). The use and misuse of V_c max in Earth system models. *Photosynthesis Research*, 119, 15–29. <https://doi.org/10.1007/s11120-013-9818-1>

4. Rezende, L. F. C., Arenque, B. C., Aidar, S. T., Moura, M. S. B., Von Randow, C., Tourigny, E., et al. (2015). Evolution and challenges of dynamic global vegetation models for some aspects of plant physiology and elevated atmospheric CO₂. *International Journal of Biometeorology*, 60(7), 945–955. <https://doi.org/10.1007/s00484-015-1087-6>

5. Bonan, G. B., Oleson, K. W., Fisher, R. A., Lasslop, G., & Reichstein, M. (2012). Reconciling leaf physiological traits and canopy flux data use of the TRY and FLUXNET databases in the Community Land Model version 4. *Journal of Geophysical Research*, 117(25C), 1–19. <https://doi.org/10.1029/2011JG001913>

They argue that FLORA having only functions and variables runs quite quickly. Despite 541 million years in the simulation, on line 133, the authors say climate model for 22 time points over the Phanerozoic – which is a small number of points for such a large period. Is that what I understood? And with a low number of times, it doesn't seem to require a lot of computational resources. The snapshots used in the model spans 541 million years. The timepoints depend on paleoclimate data that is available. It also runs into steady state at the moment meaning that the biomass output we get are for those specific snapshots only. We have to interpret the transition between each. Large computational resource is only required with the sensitivity analysis.

Yes this is correct, this paper shows model snapshots for 22 time points, and running each individual model to steady state takes only a minute or so. Nevertheless, developing and testing the model has required thousands of runs, so computational speed was still important. We have added that these runs are 'snapshots' (line 174). We will be performing continuous runs over geological time in future work.

Yes, I strongly suggest that in the future, the runnings are continuous that in this way, they can better represent the transitions of vegetation over time.

5. Not every reader has in mind memorized Earth's geological periods and plant characteristics for each period. I think it would greatly enrich the paper if information such as “The earliest land plants (embryophytes) are first identified in the Ordovician” or “Paleozoic, terrestrial flora diversified with vascular plants (tracheophytes)” could be synthesized in a table that has columns of time (560 MA, 450 MA...), from the Phanerozoic periods (Ordovician, Devonian, Carboniferous, Jurassic, Cretaceous, Modern), from the plant (embryophytes, angiosperms), and main characteristic of this plant (examples: vascular plants,).

Figures 1 and 5 with the inclusion of periods: Ordovician, Devonian... were very good, much more pleasant for readers.

6. An important point is the figures that have a low quality. The delineation of the surfaces in the maps of land are very tenuous, weak, cloudy, poorly defined, they should be sharper and could also contain more information for the reader.

Our apologies on the poor quality conversion to pdf. We have used a different converter in the revision.

The remade figures are good, much better than in the previous version.

For example, I think Figure 5 is very important in this article. However, I think it could contain more information. One of the information would be to add in the x axis of time, the periods: Ordovician, Devonian etc...

We have added the full period names to all figures including time.

OK. It's really good.

In the same Figure 5, there could be one more box with the biomass results for each of the Plant Functional Types (PFTs) defined in the simulation: tropical, boreal and temperate.

As per this and a previous reviewer comment, we have added an extra figure for the different PFT biomass overtime and some discussion (line 206; Figure 7).

Important information also requested by the other reviewers. The results show consistency in which the productivity values for tropical PFT are much higher than for PFTs: temperate and boreal. I think that this information is not the central point of the article, but it was necessary to have been presented: it enriches the article and satisfies the curiosity of the reader.

7. Still, the authors cite a lot of NPP, but do not show results. Could have a figure for NPP, similar to Figure 5 for biomass in Supplementary Information. Ordovician, Devonian etc...

We have included an NPP version of the figure in the SI (Figure S2). It follows the same trends as biomass.

Okay, it's information that also complements the article.

8. In conclusion:

In lines 207-208, when the authors say:

“In Earth’s past, these factors have changed markedly due to the positioning of the continents and changes in radiative forcing”.

The simulations with FLORA did not represent the solar variation (standardized insolation curve peaking at 400 W/m² at the equator – line 111), so I think this sentence in the conclusion is inappropriate and confuses the reader. It could be in the introduction, but warning that for simplification it was not included in the simulation.

We have now added explicit changes in solar insolation as it was mentioned by two reviewers. Equations 2-3 implement the decline of radiative forcing for times in the past. Results have also been changed to reflect this although the change is not very large as insolation only decreases by 4.6% during the Phanerozoic.

The inclusions of variations in solar radiation requested by another reviewer were good. Even though the biomass differences have not been large, it is a good refinement to the manuscript.

Reviewer #1 (Remarks to the Author):

The authors did a great job replying to the comments and improving the manuscript. I thus, recommend publication.

We thank the reviewer for their assessment.

Reviewer #2 (Remarks to the Author):

The authors have addressed my comments and those of the other reviewers from the first round of reviews. Shortcomings and incomplete descriptions from the initial draft have been remediated.

Therefore, I support the publication of this manuscript, under the condition that a few minor issues in the new version are addressed. Specifically, the following passages need to be improved:

I. 54: Capitalisation of “Early Carboniferous” is a spelling mistake, I assume.

Yes, the capital has been removed.

I. 60 – 61: This sentence is misleading! The cited paper states: “early in their evolutionary history, angiosperms could already account for a majority of the species while representing only a small proportion of the ground cover (Wing and Boucher 1998; Wing et al. 1993, 2012), and this is still the case in modern boreal forests, where angiosperms make up at least 90% of the species despite the obvious ecological dominance of conifers (Mutke and Barthlott 2005).”

If you want to bring an example for the success of angiosperm species, it makes much more sense to reference the tropics or the temperate regions, as boreal forests are commonly dominated by conifers.

We have changed the reference to a neotropics example now where angiosperms are ecologically dominant.

I. 257: Do you mean that the late Cretaceous increase in potential biomass would have facilitated angiosperm radiation? Phrase more clearly, as now it sounds like the increase may have been caused by angiosperm radiation, which, I presume, cannot be the case, as the model does not know that angiosperms radiated?

Yes, the reviewer is spot on with their interpretation. Sorry for the confusion. Changed to “may have facilitated the later radiation of angiosperm”. (Line 263)

Figure caption 7: I appreciate that you included a latitudinal analysis. However, the caption describes a different figure.

We regret the confusion here, as requested, this figure displays functional types not latitudinal locations. To clear this up we have changed “Tropical biomass” to “Tropical plant functional type biomass”.

Reviewer #3 (Remarks to the Author):

The article is quite interesting and the reading is attractive, as I commented in the first review. The authors fulfilled most of my requests, which greatly improved the manuscript. Some questions that could give more robustness to the results such as run step by step in time, and seek to better represent the transition of vegetation, the article does not contemplate, which is as a suggestion for future works. The comments to the authors' responses follow in an attached file.

We copy the file below and respond after the red statements in blue.

In black, reviewer comments in the first round.

In blue, answers from the authors.

In red, reviewer comments in the third round.

1. The authors derived the Lund-Postdam-Jena DGVM (LPJ-DGVM) model, the FLORA model and ran it for 541 million years (from the Ordovician period to modern times) estimating the carbon assimilation in the photosynthesis process, the Net Primary Productivity (NPP) and biomass – only in plant leaves. One of the objectives was to test the hypothesis that paleogeography itself influenced the dissemination of plants on Earth and for this, they observed plant biomass in relation to hydrology (runoff), local temperature and the availability of area (surface).

2. The text is good, clear, didactic and attractive to the reader. The article is very interesting and this speculation about how carbon sequestration may have been in the past of the planet and how this may have influenced the climate is quite relevant.

3. However, some points need to be improved before they are ready for publication. I think that important information is still missing for the reader. We thank the reviewer for the positive assessment and have addressed the constructive points below.

We thank the reviewer for the positive assessment and have addressed the constructive points below.

4. The authors sought to validate FLORA with results from the SDGVM model coupled to the general atmospheric circulation model (HadCM3L) (Taylor et al. 2012). The ideal would be to run more than one model (DGVM), in addition to FLORA, to give more robustness to the results. What other models out there can we run?

Much of the motivation for this work was the lack of other models available. The SDGVM is the only dynamic vegetation model we are aware of that has estimated biomass over these long timeframes. Study of the present day vegetation uses more complex models, but these tend to take much longer to run and require setting of types and locations of biomes by hand, as well as more complicated soil and ecological properties, meaning they are not well suited to predicting the spread of global biomass over deep time. We have added discussion around line 89 about the current state of the art in vegetation modelling.

The explanation of the authors is reasonable. But even so, I still think the work resents the lack of other models for comparison. We know that the models have many uncertainties, mainly in matters of plant physiology. An inaccuracy replicated over time can potentiate the error. In line 89, the authors cite references on vegetation modeling, but these references do not specifically address the uncertainties that occur in DGVMS. Since FLORA, a model derived from LPJ-DGVM, I think it is important that the authors mention that they are aware of these uncertainties and quote more appropriate references to this theme.

Suggestions for references on the uncertainties of the models:

1. Belinda EM et al. (2015). Using ecosystem experiments to improve vegetation models. *Nature Climate Change*. Vol. 5 - June 2015 – doi: 10.1038.
2. Rezende, Luiz F. C. et al., 2022. Impacts of Land Use Change and Atmospheric CO₂ on Gross Primary Productivity (GPP), Evaporation, and Climate in Southern Amazon. *Journal of Geophysical Research: Atmospheres* 10.1029/2021JD034608
3. Rogers, A. (2014). The use and misuse of Vc max in Earth system models. *Photosynthesis Research*, 119, 15–29. <https://doi.org/10.1007/s11120-013-9818-1>
4. Rezende, L. F. C., Arenque, B. C., Aidar, S. T., Moura, M. S. B., Von Randow, C., Tourigny, E., et al. (2015). Evolution and challenges of dynamic global vegetation models for some aspects of plant physiology and elevated atmospheric CO₂. *International Journal of Biometeorology*, 60(7), 945–955. <https://doi.org/10.1007/s00484-015-1087-6>
5. Bonan, G. B., Oleson, K. W., Fisher, R. A., Lasslop, G., & Reichstein, M. (2012). Reconciling leaf physiological traits and canopy flux data use of the TRY and FLUXNET databases in the Community Land Model version 4. *Journal of Geophysical Research*, 117(25C), 1–19. <https://doi.org/10.1029/2011JG001913>

Thank you for the suggestions, we have included some of these references and noted uncertainties relating to carboxylation rate where we assess our model against global data (line 155). As our model is an even more simplified version of the LPJ, the carboxylation rate seems to be the most culpable uncertainty for our calculation of plant productivity.

They argue that FLORA having only functions and variables runs quite quickly. Despite 541 Million years in the simulation, on line 133, the authors say climate model for 22 time points over the Phanerozoic – which is a small number of points for such a large period. Is that what I understood? And with a low number of times, it doesn't seem to require a lot of computational resources. The snapshots used in the model spans 541 million years. The timepoints depend on paleoclimate data that is available. It also runs into steady state at the moment meaning that the biomass output we get are for those specific snapshots only. We have to interpret the transition between each. Large computational resource is only required with the sensitivity analysis.

Yes this is correct, this paper shows model snapshots for 22 time points, and running each individual model to steady state takes only a minute or so. Nevertheless, developing and testing the model has required thousands of runs, so computational speed was still important. We have added that these runs are 'snapshots' (line 174). We will be performing continuous runs over geological time in future work.

Yes, I strongly suggest that in the future, the runnings are continuous that in this way, they can better represent the transitions of vegetation over time.

We agree on this point.

5. Not every reader has in mind memorized Earth's geological periods and plant characteristics for each period. I think it would greatly enrich the paper if information such as “The earliest land plants (embryophytes) are first identified in the Ordovician” or “Paleozoic, terrestrial flora diversified with vascular plants (tracheophytes)” could be synthesized in a table that has columns of time (560 MA, 450 MA...), from the Phanerozoic periods (Ordovician, Devonian, Carboniferous, Jurassic, Cretaceous, Modern), from the plant (embryophytes, angiosperms), and main characteristic of this plant (examples: vascular plants,).

Figures 1 and 5 with the inclusion of periods: Ordovician, Devonian... were very good, much more pleasant for readers.

Thank you.

6. An important point is the figures that have a low quality. The delineation of the surfaces in the maps of land are very tenuous, weak, cloudy, poorly defined, they should be sharper and could also contain more information for the reader.

Our apologies on the poor quality conversion to pdf. We have used a different converter in the revision.

The remade figures are good, much better than in the previous version.

Thank you.

For example, I think Figure 5 is very important in this article. However, I think it could contain more information. One of the information would be to add in the x axis of time, the periods: Ordovician, Devonian etc...

We have added the full period names to all figures including time.

OK. It's really good.

Thank you.

In the same Figure 5, there could be one more box with the biomass results for each of the Plant Functional Types (PFTs) defined in the simulation: tropical, boreal and temperate.

As per this and a previous reviewer comment, we have added an extra figure for the different PFT biomass over time and some discussion (line 206; Figure 7).

Important information also requested by the other reviewers. The results show consistency in which the productivity values for tropical PFT are much higher than for PFTs: temperate and boreal. I think that this information is not the central point of the article, but it was necessary to have been presented: it enriches the article and satisfies the curiosity of the reader.

Thank you, we also agree on this point.

7. Still, the authors cite a lot of NPP, but do not show results. Could have a figure for NPP, similar to Figure 5 for biomass in Supplementary Information. Ordovician, Devonian etc...

We have included an NPP version of the figure in the SI (Figure S2). It follows the same trends as biomass.

Okay, it's information that also complements the article.

Thank you for suggesting the figure.

8. In conclusion: In lines 207-208, when the authors say:

“In Earth’s past, these factors have changed markedly due to the positioning of the continents and changes in radiative forcing”. The simulations with FLORA did not represent the solar variation (standardized insolation curve peaking at 400 W/m² at the equator – line 111), so I

think this sentence in the conclusion is inappropriate and confuses the reader. It could be in the introduction, but warning that for simplification it was not included in the simulation.

We have now added explicit changes in solar insolation as it was mentioned by two reviewers. Equations 2-3 implement the decline of radiative forcing for times in the past. Results have also been changed to reflect this although the change is not very large as insolation only decreases by 4.6% during the Phanerozoic.

The inclusions of variations in solar radiation requested by another reviewer were good. Even though the biomass differences have not been large, it is a good refinement to the manuscript.

Thank you.